# Hail Climatology in the Mediterranean Basin Using the GPM Constellation (1999–2021)

**Sante Laviola** *[iD], **Giulio Monte, Elsa Cattani** [iD] **and Vincenzo Levizzani** [iD]

CNR-ISAC, Via Gobetti 101, 40129 Bologna, Italy
* Correspondence: s.laviola@isac.cnr.it; Tel.: +39-051-639-8019

**Abstract:** The impacts of hailstorms on human beings and structures and the associated high economic costs have raised significant interest in studying storm mechanisms and climatology, thus producing a substantial amount of literature in the field. To contribute to this effort, we have explored the hail frequency in the Mediterranean basin during the last two decades (1999–2021) on the basis of hail occurrences derived from the observations of the microwave radiometers on board satellites of the Global Precipitation Measurement Constellation (GPM-C) from 2014 (date of GPM Core Observatory launch) onwards and merging multiple other satellite platforms prior to 2014. According to the MWCC-H method, two hail event categories (hail and super hail) are identified, and their spatiotemporal distributions are evaluated to identify the hail development areas in the Mediterranean and the corresponding monthly climatology of hail occurrences. Our results show that the northern sectors of the domain (France, Alpine Region, Po Valley, and Central-Eastern Europe) tend to be hit by hailstorms from June to August, while the central sectors (from Spain to Turkey) are more affected as autumn approaches. The trend analysis shows that the mean number of hail events over the entire domain tends to substantially increase, showing a higher increment during 2010–2021 than during 1999–2010. This behavior was particularly enhanced over Southern Italy and the Balkans. Our findings point to the existence of "sub-hotspots", i.e., Mediterranean regions most susceptible to hail events and thus possibly more vulnerable to climate change effects.

**Keywords:** hailstorms; climatology; Mediterranean; GPM-C passive microwaves

## 1. Introduction

Hailstorms are among the most impactful phenomena on infrastructures and human activities. The impacted sectors depend on the hail size, with damages in agriculture occurring even in the presence of small size hail and serious damage to vehicles and buildings caused by hailstones with large to very large diameters. The Mediterranean basin is well-known to be a vulnerable planetary climate hotspot that is affected by hailfall episodes potentially affected by climate change.

Several studies have analyzed the distribution of hail events among Mediterranean countries by tracing the frequency of hailstorms and inspecting the related social impacts and damages. In their exhaustive review, Punge and Kunz [1] described the variation of hail frequency over the European countries, where the increasing severity of hailstorms is suggested to be associated with climate change. Although the connection between hailstorm trends and climate change remains difficult to establish unambiguously, several indicators seem to link the observations of increasing damages from hail-producing storms to a climate change perspective.

Sanchez et al. [2] analyzed the National Center for Environmental Prediction (NCEP) reanalyses for the period 1948–2015 to evaluate the variation of the atmospheric conditions favoring hail formation in southern France. The effects of climate change on hailstorms are also discussed in Raupach et al. [3]. Here, the main climate variables are used to describe

the mechanisms of hail formation and growth by inferring the future modifications due to the warming climate for several countries.

As described by Prein and Holland [4] and Tippett et al. [5], the limited capacity of climate models in simulating hailstorms and the sparsity of historical data records are the main limitations to understand the interaction between the climate system and severe convection. Satellite remote sensing offers a favored observational perspective for detecting convective clouds potentially forming hailstones. Satellite multi-channel measurements of cloud top brightness temperatures are exploited to recognize the overshooting plumes often associated with severe convection [6–11]. Although the presence of overshooting cloud tops (OTs) tends to induce an overestimation of the frequency of hailstorms, this is a robust approach to distinguish potential hail-bearing clouds. Punge et al. [12] combined the OT observations from the Meteosat Second Generation (MSG) satellite with a hail-specific filter derived from the ERA-INTERIM reanalysis to obtain hail frequency in Europe and identify regions highly exposed to hail events. Microwave radiometers provide a more direct measurement of the signal attenuation by frozen hydrometeors from within the storms, while being characterized by a temporal discontinuity due to their orbital characteristics. Using the frequencies at 85, 37, and 19 GHz on board the Tropical Rainfall Measurement Mission (TRMM), Cecil [13] inferred the distribution of intense thunderstorms connected with the presence of large hail by relating the brightness temperature reduction to the hail diameters derived from ground hail reports. Recent works extended the sounding frequencies to 90 GHz to increase the sensitivity to smaller hailstones as a precursory condition for the growth of damaging hail (d > 20 mm). Ferraro et al. [14] developed a pioneering prototype method based on high frequency channels on board the Advanced Microwave Sounding Unit-B (AMSU-B).

This work has its premises in the results presented in Laviola et al. [15,16]. Laviola et al. [15] devised a new modified logistic model for the relation between the scattering signal in the microwaves and the hail diameter by varying the hail cross-section; the model quantifies the extinction of radiation due to hailstones and ice aggregates through the application of a probabilistic growth model. The model was applied to the satellite platforms in orbit and validated against 12 years of surface observations. The method was then extended to the Global Precipitation Measurement Constellation (GPM-C) [16] to account for instrumental differences among the various microwave radiometers and attain a global observation perspective at high temporal rate. Strong performances were demonstrated in the detection of different hail-bearing storms and seasonality of the events. The present study is the first that fully exploits the potential of cloud-penetrating satellite microwave radiometer frequencies to derive the occurrences and the long-term (1999–2021) spatiotemporal variability of hail events on the Mediterranean basin based on the potential of the probabilistic method using the GPM-C. The study provides a long and consistent dataset over the area based on physical cloud observations using all radiometers in orbit, extending previous studies based on a single radiometer and conducted over a shorter period [17]. In fact, the main reason that justifies the study is producing a new climatology of hail episodes affecting the Mediterranean domain. The focus is on the characterization of hail events in terms of spatial distribution, monthly climatology and trends.

The hail dataset exploited in the present analysis is derived from the probability-based MicroWave Cloud Classification-Hail (MWCC-H) [15,16] mathod applied to frequency channels in the range 150–170 GHz of the GPM-C and modified for applications to the sensors available in the pre-GPM era. The MWCC-H method associates probability values to microwave signals from very small ice particles (low probability) while very large hailstones (d > 10 cm) are typically marked by likelihoods close to 1. Three hail categories are identified by MWCC-H (i.e., Graupel/Hail Initiation (HI) with d < 2 cm; Large Hail (H) with 2 < d < 10 cm; Super Hail (SH) with d > 10 cm), associating the brightness temperatures (BT) values with hail probability ranges (i.e., [0.36, 0.45], [0.45, 0.60] and >0.60 for Hail Initiation, large Hail, and Super Hail, respectively). A new category covering the hail

probability range [0.20, 0.36] has been recently introduced: Hail potential (HP) identifies potential hail-bearing convection with no hail detected.

The consistency of the hail dataset allowed for the evaluation of the seasonal characteristics of hailstorms focusing on the most affected areas of the Mediterranean basin. Our results demonstrate that the dataset derived from the MWCC-H method well-describes the trend of hail events as predicted from the main climatic indicators [5].

Section 2 presents the characteristics of the study area. The hail data collection and analysis methods are illustrated in Section 3. Section 4 describes the results in terms of monthly climatology and trends to identify the Mediterranean areas mostly affected by hailstorms. Results are discussed in Section 5. Finally, possible expectation for future hailstorm events in the context of climate change are discussed in Section 6.

## 2. The Study Area

A high number of intense cyclones occur every year in the Mediterranean basin, producing a broad range of severe socio-economic and environmental impacts in this densely populated region. Recent examples of such events are the extra-tropical cyclones producing hail that hit the region, Greece and Italy in particular, in October–November 2021. Actually, 2021 was a very peculiar year for severe hailstorms across Europe, with 5195 reports of large hail ($\geq$2 cm), 871 reports of very large hail ($\geq$5 cm), and 29 reports of giant hail ($\geq$10 cm) registered in the European Severe Weather Database (ESWD) for the period January–October 2021 (https://www.essl.org/cms/hailstorms-of-2021/, accessed on 24 March 2022).

The Mediterranean basin is the largest of the semi-enclosed European seas with a coastline of 46,000 km, where approximately one third of the population (about 480 million people) is concentrated. Cyclogenesis is exceptionally frequent [18], influenced by the complex topographical features, as well as the unique location between the tropics and the mid-latitudes, downstream of the North Atlantic storm tracks. Mediterranean cyclones often have a high impact, being responsible for the great majority of precipitation and wind extremes in the area [19].

The analyzed area is presented in Figure 1. The spatial domain has been divided into nine geographical sectors to quantify the impact of hailstorms more locally and possibly identify the areas more affected by hail.

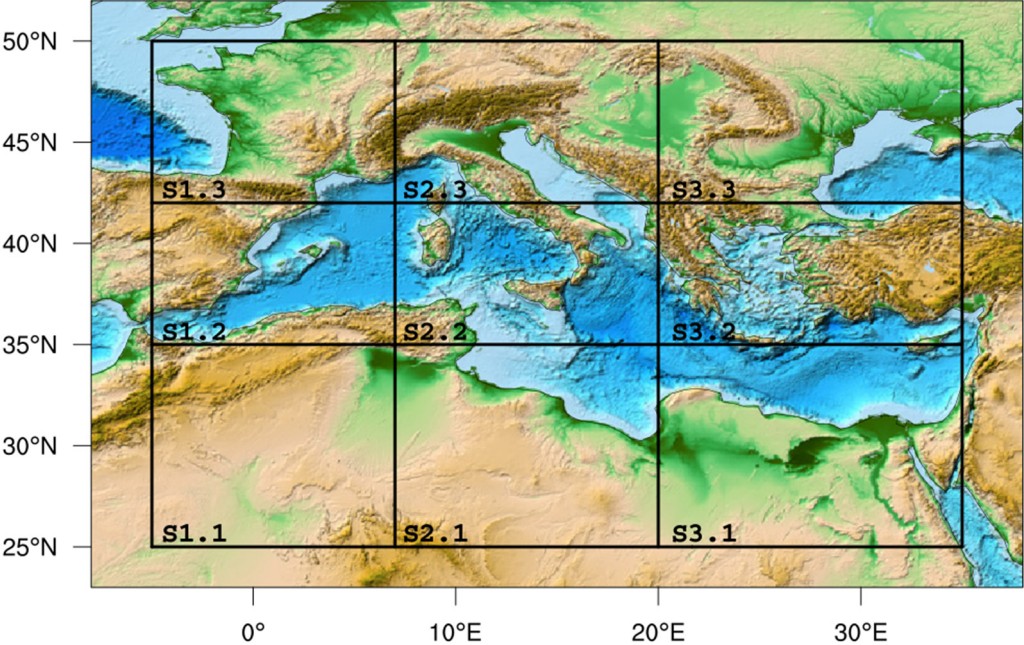

**Figure 1.** The study area and the nine sectors used for the analysis.

## 3. Materials and Methods

### 3.1. Hail Data Collection

Hail data used in this study were retrieved from the microwave sensors' observations at high frequency acquired during the last 22 years (1999–2021). The Advanced Microwave Sounding Unit-B (AMSU-B) launched on board the National Oceanic and Atmospheric Administration-15 (N15) satellite in 1999 opened the exploitation of high spatial resolution data at high-frequency channels for the identification of the hail signature in intense thunderstorms [15]. From then on, the data availability increased progressively with further satellites with sensors operating AMSU-B like channels up to the GPM mission. The temporal sequence of the satellites operational during the period 1999–2021 is presented in Figure 2. Although the first two years suffer from scarcity of data, from 2001, the increasing number of AMSU-B/Microwave Humidity Sounding (hereafter MHS for both sensors) twin sensors onboard N16, N17, N18, N19 and Metop-A, -B, and -C together with the Special Sensor Microwave-Imager/Sounder (SSMIS) onboard F-16 and 17 significantly improved the observation rate. In 2014, the new GPM Microwave Imager (GMI) with the whole GPM-C further enhanced the global observations of precipitation by providing more detailed information on the evolution of storms. Laviola et al. [16] demonstrated the capability of the GPM-C to detect hail providing hail maps at high temporal resolution to monitor the evolution of hail-bearing clouds.

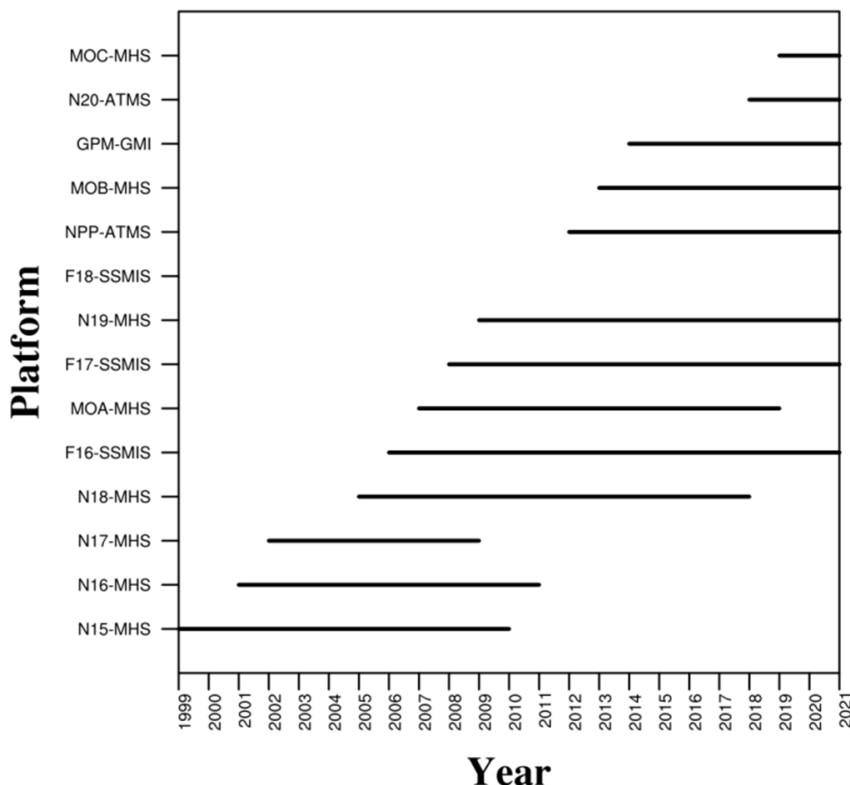

**Figure 2.** Operational satellite platforms between 1999 and 2021.

In Table 1, we show the four hail categories from the probability-based method MWCC-H and the associated potential severity. Note that the increase of the hail probability as a function of hail diameter is associated with an increase of the kinetic energy (computed with the fourth power of the radius [20]) and in turn of the severity of the event.

**Table 1.** MWCC-H hail categories as a function of the kinetic energy and potential severity at the ground.

| Category Description | Probability of Hail | Diameter Range (cm) | Kinetic Energy (J) | Terminal Velocity (m s$^{-1}$) | Potential Severity |
|---|---|---|---|---|---|
| Hail Potential (HP) | 0.20 ÷ 0.36 | ~ | ~ | ~ | Absent to low |
| Graupel/Hail Initiation (HI) | 0.36 ÷ 0.45 | <2 | <33.84 × 10$^{-2}$ | <19.09 | Low to moderate |
| Large Hail (H) | 0.45 ÷ 0.60 | 2 ÷ 10 | 33.84 × 10$^{-2}$ ÷ 423 | 19.09 ÷ 42.69 | High to severe |
| Super Hail (SH) | > 0.60 | >10 | >423 | >42.69 | Severe to extreme |

The four categories are introduced to describe the characteristics of the hailstones from very small to graupel and finally large and very large hail. These four categories are instrumental to fully describe the dynamics of hailstorms, but in this study only categories associated with the most severe events were selected. Thus, for this analysis, the method MWCC-H was used to identify satellite pixels with Hail (H, hailstone diameter $2 < d < 10$ cm) or Super Hail (SH, $d > 10$ cm). The second hail category identified by the method (Graupel/Hail Initiation with $d < 2$ cm) was not considered for the present study. The MWCC-H was applied separately to each sensor/platform dataset of Figure 2. The H and SH occurrences (i.e., number of pixels) have subsequently been layered for each month of the considered period and each geographical sector.

The exploitation of the various GPM-C sensors implied the adjustment of the sensor channels in the range 150–170 GHz to reduce possible radiometric discrepancies between the reference instrument MHS, for which MWCC-H was originally conceived, and the MHS-like sensors of the GPM-C according to the procedure presented in [16]. Moreover, the limb effect was considered for cross-track sensors to reduce the possible signal overestimations due to the observation angle.

For the cross-track sensors of the constellation used in this study, the size of the Instantaneous field of view (IFOV) tends to increase moving from the nadir position to the edges of the swath so that the spatial resolution degrades correspondingly. A coarser horizontal resolution has a negative impact on the estimation of the hail events for two reasons: (1) hailstorms generally belong to convective systems affecting limited areas so high-resolution observations are required, and (2) the increased slant path in the atmosphere can further reduce cloud ice particle radiance reaching the sensor, leading to a depression of the microwave signal that may wrongly be attributed to the presence of hail. Several methods were developed to correct AMSU-B, MHS, and the Advanced Technology Microwave Sounders (ATMS) measurements for the limb effect [21,22]. However, in this study a conservative, but probably more effective, approach was adopted considering only the central portion of the swath line and neglecting 20 IFOVs at both scan line edges. Thus, only the central pixels of the swath are retained to preserve the higher spatial resolution at the ground around the nadir position: from about 8–10 km of GMI to 14 km of SSMIS and 16–25 km of MHS/ATMS. In this context, an H or SH event means a satellite pixel where MWCC-H identifies the presence of the H or SH categories. Nevertheless, due to the spatial resolution of the satellite data, generally H- and SH-storms can reasonably cover about 5–6 or 1–2 pixels according to the previously mentioned spatial resolutions. However, the spatial reconstruction of the single H or SH storms is not carried out and each pixel composing the hail system is retained separately. The number of the processed data and H and SH events as a function of the year are shown in Figure 3.

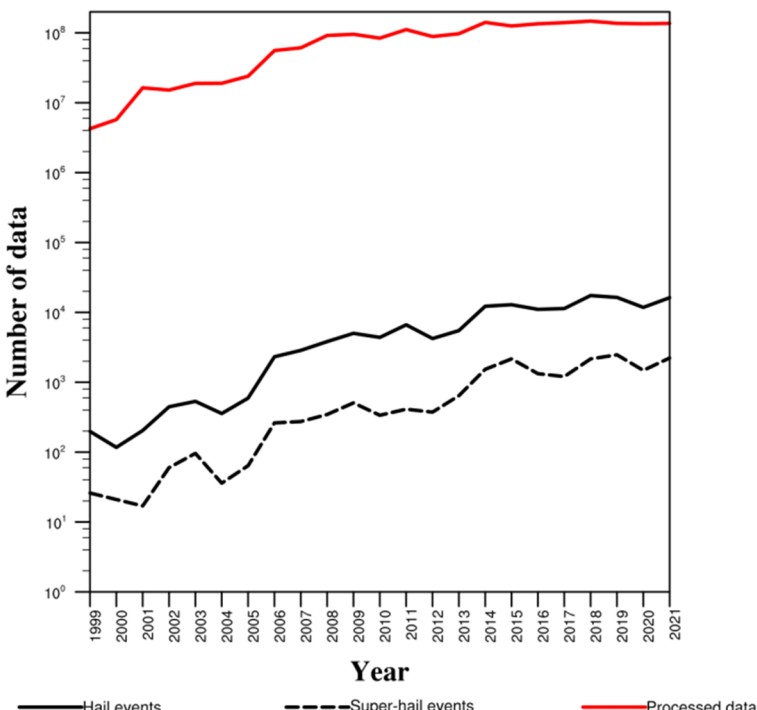

**Figure 3.** Number of the processed satellite pixels for each year (red line), hail events (solid black line), and super hail events (dashed black line).

For the present analysis, only H and SH events occurring from April to November during the period 1999–2021 were considered. The so-called "hail season" is the most hail-affected period of the year in all sectors of the domain [23]. The winter months (December, January and February) and the beginning of spring (March) were discarded, mainly for the following reasons:

1. These months are characterized by the lowest probability of experiencing hailstorms according to the results of the MWCC-H method.
2. In clear-sky conditions, the presence of snow on the ground often observed in wintertime can generate a MW radiometric signal similar to that of cloud-ice particles, thus generating a misleading response in terms of H or SH occurrences.

*3.2. Trend Analysis*

The presence of significant monotonic trends in the annual time series of H and SH event occurrences is assessed with the Mann–Kendall (M-K) test, whereas the estimate of the trend rates is performed with the Theil-Sen estimator [24–26]. The null hypothesis, i.e., the absence of monotonic trends, is rejected when a significance level of at least 10% is reached. The trend analysis was implemented with the function "trend_manken" of the NCAR Command Language software package [27], applicable to time series characterized by evenly spaced values and composed at least by 10 values.

Trends were assessed for the Mediterranean basin as a whole and for each sector using time series of the annual number of H or SH events (i.e., considering the hail season April-November). To prevent the effects on trends of the increasing number of available observations with time (Figure 3) and the variable sensor spatial resolution, the trend evaluation was performed using time series assembled by collecting observations according to the following criteria:

1. link different platforms to cover the whole study period 1999–2021 with a single operational platform at a time;
2. exploit sensors with the same spatial resolution, i.e., MHS and ATMS.

Three time series were analyzed to verify the robustness and repeatability of the results, whose characteristics are summarized in Table 2.

**Table 2.** Composition of the analyzed time series of annual number of H or SH events.

| 1. Time Series N15-N18-NPP | | |
| --- | --- | --- |
| N15 | N18 | NPP |
| 01/04/1999 31/05/2005 | 01/06/2005 30/11/2011 | 01/04/2012 30/11/2021 |
| **2. Time Series N15-N18-MOB** | | |
| N15 | N18 | MOB |
| 01/04/1999 31/05/2005 | 01/06/2005 30/11/2012 | 01/04/2013 30/11/2021 |
| **3. Time Series N15-MOA-MOC** | | |
| N15 | MOA | MOC |
| 01/04/1999 31/05/2007 | 01/06/2007 30/11/2019 | 01/04/2020 30/11/2021 |

## 4. Results

### 4.1. Monthly Climatology

In Figure 4 the monthly distributions of H and SH events that occurred in the Mediterranean basin are reported. All the results discussed in this section are obtained using all satellites available during the period 1999–2021. Each analyzed month was subject to the same increase in the number of observations (Figure 3), thus preserving the shape of the H and SH event monthly cycles and permitting the intercomparisons of the monthly cycles from the various sectors. H events reach their maximum in September and October (about 28,783 and 29,508 H occurrences, respectively) with a secondary peak in June (20,498). SH events are much more concentrated during the period from August to November with again a maximum in September and October (5350 and 4731, respectively). April and May do not show a significant number of events.

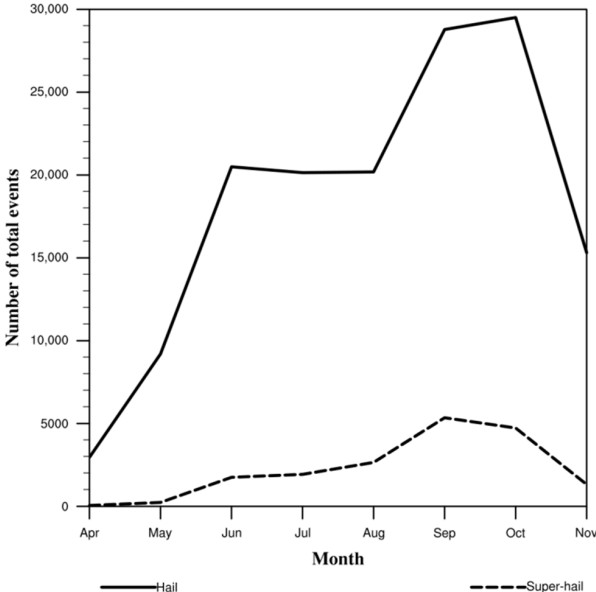

**Figure 4.** Monthly distributions of hail (solid line) and super-hail (dashed line) events occurred in the Mediterranean basin in the period April-November from 1999 to 2021.

The total numbers of H and SH events that occurred in each month of the hail season and each geographical sector for the whole period 1999–2021 are illustrated in

Figures 5 and 6. The monthly occurrence of the H and SH events in each sector is coherent with the distribution of hail calculated for the whole Mediterranean basin (Figure 4), but provides also further insights concerning the spatial distribution of the hail seasonality. A distinct subdivision of the studied area with respect to the hail peak month is evident. Northern sectors (S1.3, S2.3, and S3.3) reach the H event maxima during summer (July, July, and June, respectively), with SH event maxima occurring in September, and again July and June. The H and SH event maxima occur between September and October in the other sectors. Sectors 2.2 and 2.3, framing the Italian peninsula, and central Mediterranean, are the most affected by H events with 32,152 and 28,986 cases, and by SH events with 6051 and 3237 cases, respectively.

The main characteristics of the H and SH seasonality can be summarized as follows:

1.  April is the month with the lowest number of occurrences, having S1.1 and S1.3 (Algerian basin, Spain, France, and South Belgium) the maximum number of H (571) and SH (14) with respect to the other sectors, respectively. In this month the SH events are almost absent.
2.  From May to August, H events affect mainly the northern regions of the domain, in particular sectors S2.3 (Northern Italy, Alps Region, and Dinaric Alps) with the highest peak in July and August, and S3.3 (Balkans and Carpathians mountains) with highest peaks in June and July with respect to other sectors; H occurrences in S1.3 (France and Southern Belgium) is lower than those of the previous sectors. From a dynamic point of view, the distribution of H events could be explained by considering the shift towards higher latitudes that Atlantic storm tracks experience during summertime, when the Iberian Peninsula, Southern Italy, and the southern part of the Balkans are generally protected by a subtropical anticyclonic belt. A similar spatial distribution is valid also for the SH events that reach their maxima in July and August for S2.3 (928 and 815 events, respectively) and in May and June for S3.3 (89 and 777, respectively). The central sectors (S1.2, S2.2 and S3.2) had lower occurrences compared to the northern areas of the domain. Finally, H and SH events in the southern sectors of the domain (North Africa and Middle East) were negligible or even null.

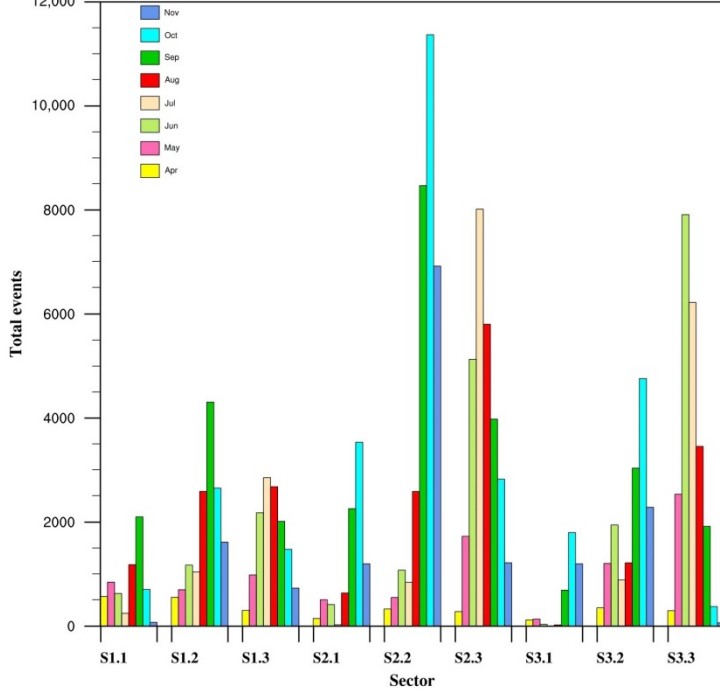

**Figure 5.** Total number of H events in the period April-November from 1999 to 2021 for each sector of Figure 1.

3. In autumn (September to November), the highest number of H events are found in the central sectors of the domain. In particular, S2.2 (Southern Italy and Tunisia) is the sector with the highest number of H pixels in each month of the period (8496, 11,369, and 6921). The sector S1.2 (Spain, Balearic Islands and Western Mediterranean) showed its highest number of H occurrences in September (4308), while S3.2 (Greece and the Aegean Sea) experienced its highest number of H events in October (4761). Such a distribution of hail events in this period of the year could be supported by the action of the polar jet stream; as autumn approaches, a secondary branch of the jet stream often reaches the western part of the Mediterranean Sea enhancing the development of troughs that generally affect the whole basin in the W-E direction. These systems are fed along their path by humidity produced by high sea surface temperatures and can sometimes trigger severe convective instability conditions. The number of H events also generally increased in the southern sectors, even though to a lesser degree, especially in S2.1 (South Mediterranean, Southern Tunisia and Libya), probably due to the presence of high sea surface temperatures in contrast with the nearby land. Though the total number of SH events was considerably lower than that of the corresponding H events, it can be noted that in some sectors the seasonality of SH events is preserved, such as in S2.2 and S2.3, which stand out again as the sectors most affected by SH events.

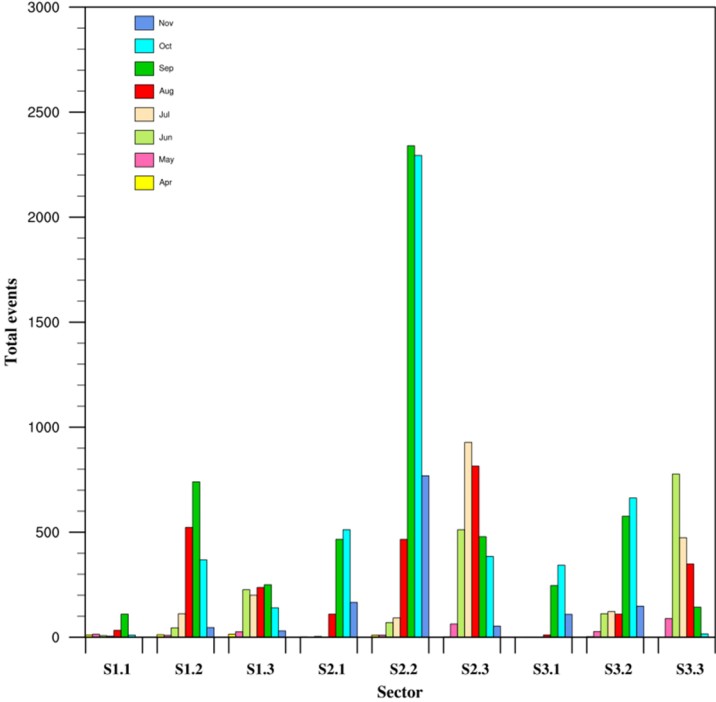

**Figure 6.** Total number of SH events in the period April-November from 1999 to 2021 for each sector of Figure 1.

### 4.2. Trend Analysis

#### 4.2.1. Trends over the Entire Mediterranean Basin

All explored time series highlighted an increasing trend in the H event occurrences with the worst significance of about 7% in the case of time series-2 (Figure 7b). A substantial increase in H events is detected since 2012–2013 as documented from the dashed red lines in Figure 7a–c, which represent the mean values of the number of H events in the two sub-periods, 1999–2010 and 2010–2021. This step-trend is particularly evident for time series 1 and 2, which are both composed by the sequence of N15 and N18 data in the first sub-period.

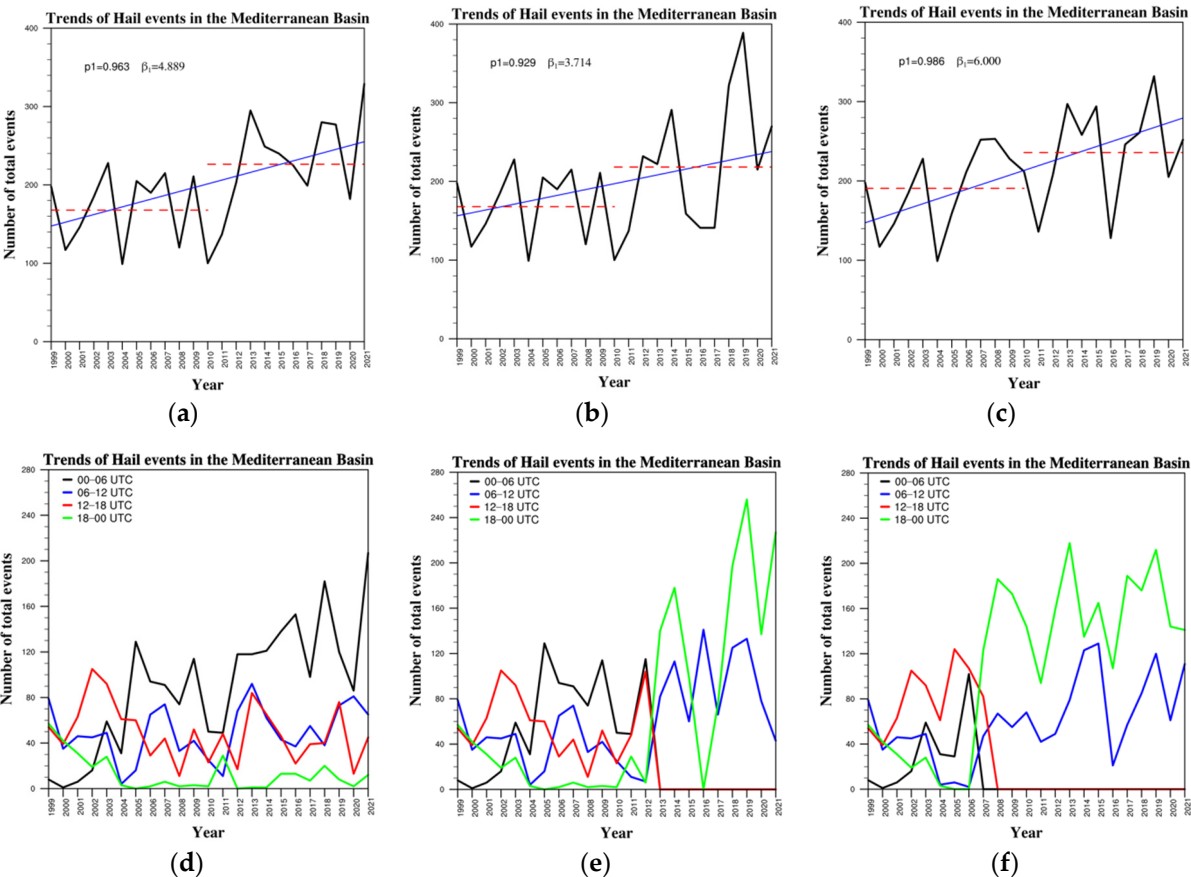

**Figure 7.** Temporal evolution of the annual number of H events for the entire Mediterranean basin considering the three-time series according to Table 2, i.e., time series-1 N15-N18-NPP (**a**,**d**), time series-2 N15-N18-MOB (**b**,**e**), and time series-3 N15-MOA-MOC (**c**,**f**). In (**a**–**c**) p1 and $\beta_1$ represent the confidence level and the trend rate (number of H events/year), respectively; red dashed lines identify the mean values of the number of H events in the two analyzed decades; blue line is the linear trend. Colored curves in (**d**–**f**) present the evolution of the annual number of H event in four temporal windows, i.e., 00–06, 06–12, 12–18, and 18–00 UTC.

In Figure 7d–f the decomposition of the total number of H events (black curves displayed in Figure 7a–c) in four temporal windows (i.e., 00–06, 06–12, 12–18, and 18–00 UTC) is shown. Each curve is obtained considering the satellite observations belonging to a specific temporal window, considering the UTC time when the satellites enter the study area. All sensors in this work are on board sun-synchronous platforms visiting the pixels of the study area twice a day at fixed local times, except for GMI on board the GMP Core Observatory (GPM-CO). Thus, it is necessary to establish whether trends (Figure 7a–c) are affected by the specific observation times of each platform modulating the observation availability. The annual numbers of processed pixels for each time series and time window are presented in Figure 8.

The positive trend of the time series 1 (Figure 7a) seems supported by the increasing number of H events in the 00–06 UTC window ($p_1 = 1$ and $\beta_1 = 6.55$ number of H events/year). The positive trend could be initially affected by the corresponding increase in the number of processed pixels (Figure 8a, black curve) (i.e., more processed observations, more hail events). The number of processed pixels in the 00–06 UTC window stabilizes after 2006, but the H events continue to increase, reaching their maximum at the end of the analyzed period. The other temporal windows do not reveal any trend despite the significant variations of the pixel number over the years, in particular for the 06–12 and 12–18 UTC time windows.

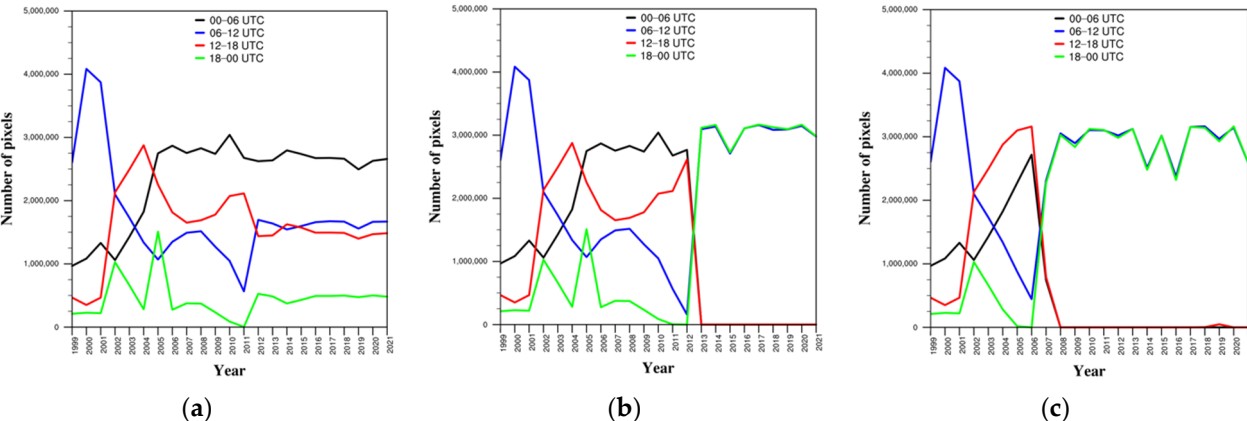

**Figure 8.** Annual number of pixels processed by the MWCC-H method in the four-time windows. Time series N15-N18-NPP (**a**); time series N15-N18-MOB (**b**); time series N15-MOA-MOC (**c**).

A similar signal from the 00–06 UTC window is not perceivable from time series 2 (Figure 7e), where MOB-MHS observations replace the NPP-ATMS ones from 2013. MOB-MHS does not provide observations in the time ranges 00–06 and 12–18 UTC (Figure 8b). Conversely, a positive trend characterizes the 06–12 UTC window ($p_1 = 0.95$ and $\beta_1 = 3.62$ number of H events/year). This fact, together with the high number of H events observed uniquely by MOB-MHS in the 18–00 UTC window (Figures 7e and 8b, green curves), supports the general positive trend. However, the results from the 06–12 UTC window are difficult to interpret considering the highly variable number of processed pixels in this time interval, where the increasing trend could be caused by the substantial rise of the MOB observations.

N18 observations are replaced by those of N15 up to 2007 and subsequently by MOA up to 2019 and MOC for the last two years in the time series 3. Thus, like in the previous case, only the 06–12 and 18–24UTC time ranges have observations for the entire period (Figure 8c). Both time intervals give rise to positive and significant trends ($p_1 = 0.98$ and $\beta_1 = 2.44$ number of H events/year, and $p_1 = 0.99$ and $\beta_1 = 8.20$ number of H events/year for 06–12 and 18–00 UTC, respectively). After a first phase characterized by high variations, starting from 2008 the number of observations in the 06–12 UTC window is relatively more stable. Nevertheless, an increasing number of H events is detected by the MWCC-H method. The H event trend mimics the variations of the number of processed pixels in the 18–00 UTC window.

Considering the SH events (Figure 9), only the time series 3 presents a positive and significant trend, which is replicated in the 06–12 and 18–00 UTC windows. The situation is analogous to the previous one described for the H events.

### 4.2.2. Trends over Sectors

H and SH event trends were also investigated in each sector considering the three satellite time series reported in Table 2 to identify the regions in the Mediterranean basin where the general positive trends previously discussed are confirmed. Only Sector 3.1, corresponding to south-eastern Mediterranean, exhibits positive trends of the number of H events for the three-time series with a confidence ≥ 91% (Figure 10). However, the similarity among the three-time series of the H events is very low.

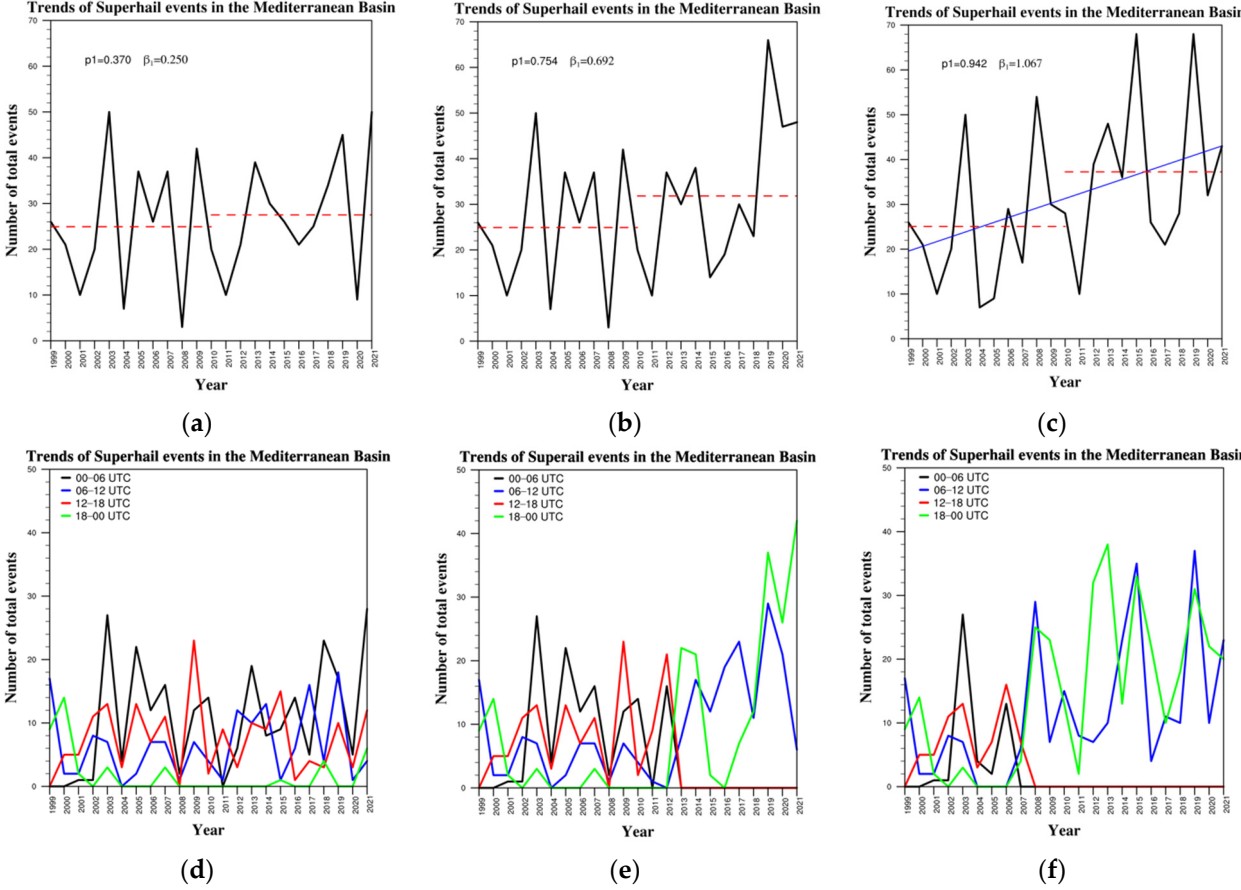

**Figure 9.** (**a**–**f**) Same as in Figure 7 but for SH events.

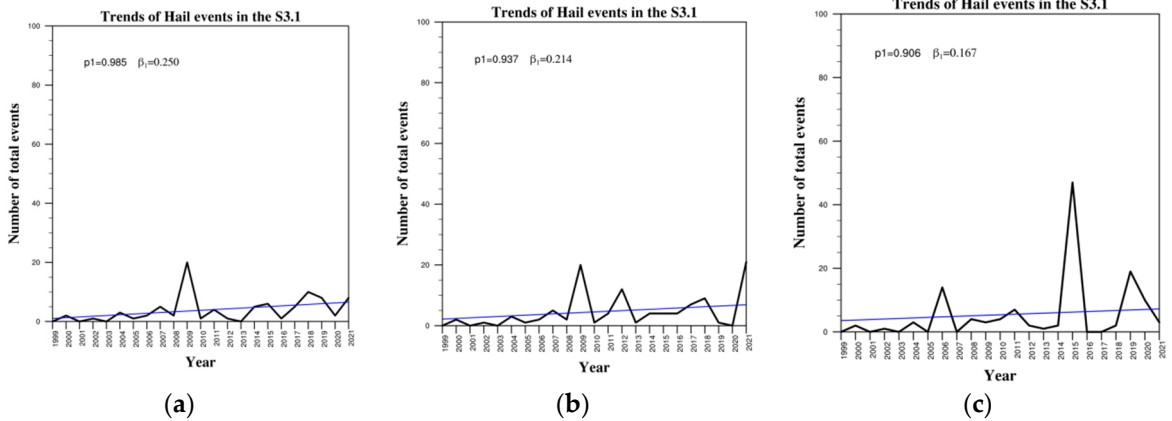

**Figure 10.** Temporal evolution of the annual number of H events for the sector 3.1 considering the three-time series in Table 2, i.e., time series-1 N15-N18-NPP (**a**), time series-2 N15-N18-MOB (**b**), and time series-3 N15-MOA-MOC (**c**).

Significant positive trends are also identified in sectors 2.1 (North Africa and Libya) and 3.2 (Greece and Turkey) (Figures 11 and 12). However, in these cases, the results are confirmed only by two time series. Results from the Libyan sector can be considered not completely reliable since hailstorms are not so frequent in this area, even though the country recently witnessed some severe events, such as the one on 24 October 2021 in north-western Libya and on 27 October 2020, when very large hail with diameter up to 20 cm hit the city of Tripoli.

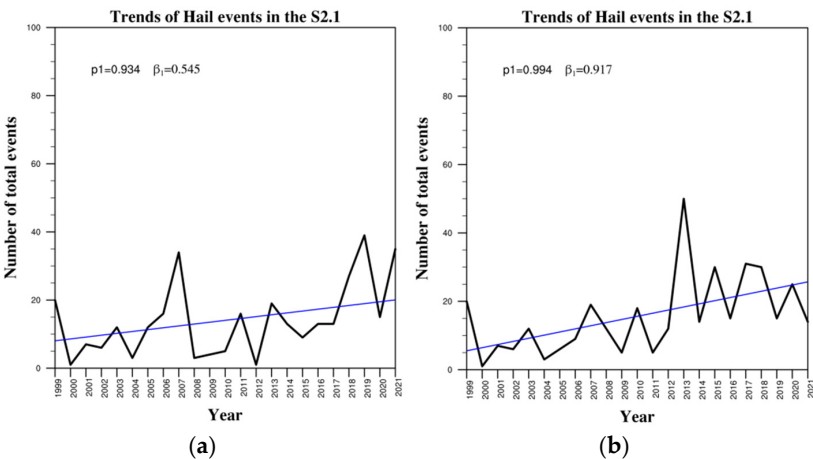

**Figure 11.** Temporal evolution of the annual number of H events for the sector 2.1 considering the time series-2 N15-N18-MOB (**a**), and time series-3 N15-MOA-MOC (**b**). Blue line displays the tendency of series.

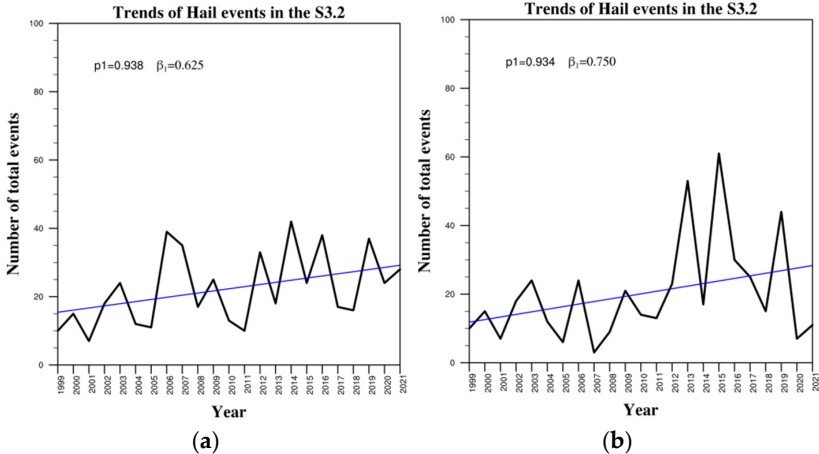

**Figure 12.** Temporal evolution of the annual number of H events for the sector 3.2 considering the time series-1 N15-N18-NPP (**a**), and time series-3 N15-MOA-MOC (**b**). Blue line displays the tendency of series.

As for SH events significant positive trends are exhibited only by time series 3 in sectors 3.1, 3.2, 2.1, and 1.2 (Figure 13).

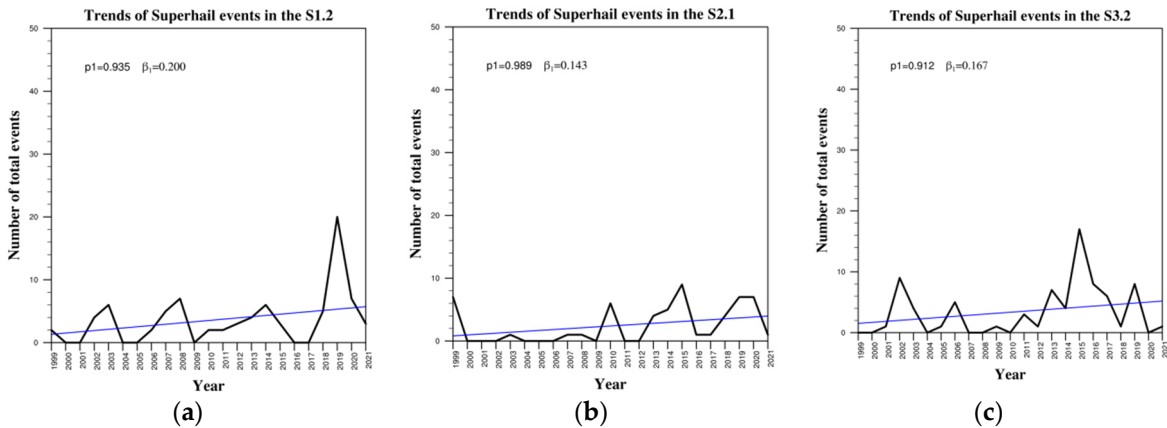

**Figure 13.** Temporal evolution of the annual number of SH events for the sectors 1.2 (**a**), 2.1 (**b**) and 3.2 (**c**) considering the time series-3 N15-MOA-MOC. Blue line displays the tendency of series.

### 4.3. Hail Event Diurnal Cycle

The annual number of H and SH events for the period 2013–2018 was investigated in each time window of the previous section to identify the time of the day more affected by hailfall incidence. Each time window was populated with H events detected using different platforms to guarantee year by year similar numbers of processed pixels. Thus, the following combinations of time windows and platforms were selected:

1.  00–06 UTC NPP-ATMS
2.  06–12 UTC MOB-MHS
3.  12–18 UTC N18-MHS
4.  18–00 UTC MOA-MHS.

Figure 14 shows the relative frequencies of H and SH events that occurred during the period 2013–2018 in the four-time intervals. H events are overall concentrated during the 00–06 and 18–00 UTC intervals, with 27% and 32% of the total number of H events occurring in these two intervals. Similar frequencies around 22% characterize the other times of the day. The number of SH events increases during the day, reaching its maximum at 18–00 UTC with 34% of the SH occurrences. The evaluation of the hail diurnal cycle was attempted also by Púčik et al. [28] over Europe, considering the large hail reports from ESWD, and Manzato [29] exploiting the hailpad network operational since 1988 in the Friuli-Venezia-Giulia region in Italy. Although very different areas were considered and the hail event definition was different, both works agree on the fact that hail events are much more frequent in the evening. In our case, this result is confirmed by SH events but not by H events. However, comparing our results with their findings, the different nature of the exploited datasets has to be considered. Hail signals identified by satellite data through the MWCC-H method relate to the presence of ice inside clouds and are associated with hail probabilities. Thus, it is possible that not all satellite-based hail pixels are characterized by hailstones reaching the ground as in the case of the ESWD data set or the hailpad network.

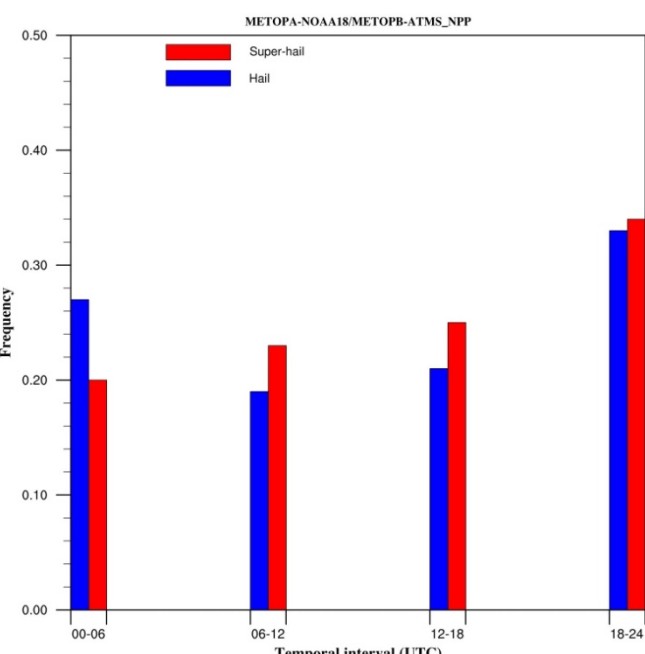

**Figure 14.** Relative frequencies of H (blue) and SH (red) events for the period 2013–2018 for the four-time intervals 00–06, 06–12, 12–18, and 18–00 UTC.

## 5. Discussion

We analyzed the last 22 years of MWCC-H-derived hail events in the Mediterranean basin. The main reason that justifies our study is producing a new climatology of hail episodes affecting a very complex domain which is well-known to be a planetary climate

hotspot, namely a region particularly vulnerable to the impact of climate change. Thus, to quantify the variability of hail impact and identify the areas susceptible to hail events, we built a long-term database on the basis of identification of the MWCC-H method. The potential of the MWCC-H method in detecting hail events arises from its capability to fit well all high-frequency microwave radiometers flying from the AMSU era (i.e., since 1999). The 12yr validation and the adjustment of the MWCC-H original scheme to all MHS-like radiometers orbiting with the GPM-C enhanced the detection skills of rapidly evolving systems such as hailstorms [15,16].

The first step of our investigation was the partitioning of the study area into nine different geographic domains to identify the Mediterranean sub-regions particularly subject to hail occurrences. Thus, the statistics were calculated to define the spatiotemporal characteristics of hail events and extract the underlying pattern of behavior in terms of numerosity, recurrence and trends of the most severe events. Our results sustain and reinforce previous studies while at the same time opening a new perspective on possible future variation of hailstorms in terms of number and intensity due to climate change.

The monthly climatology shown in Section 4.1 demonstrates that hail events occurring from late spring to summer (Jun, Jul, Aug) affect in particular the northern sectors (S1.3, S2.3, S3.3) of the study domain. The same seasonal distribution can be extended also to the rare hail events, namely SH.

Central Europe (S1.3, S2.3) is highly exposed to hail hazard, namely Switzerland, Northern Italy, Austria, and Germany, where several comprehensive studies on hail frequency are available (see [1] for a comprehensive review). It can be assumed that hail frequency decreases from west to east and from south to north. Continentality is one of the governing factors of hail frequency here, as it leads to lower moisture contents and lesser frontal systems and therefore less favorable conditions for convective activity. On the other hand, increasing insolation and decreasing wind shear from north to south contribute to the latitudinal gradient. Hail frequency is generally enhanced over the pre-Alpine regions of Switzerland, Austria, and Slovenia. Overall, the most hail-prone regions are over the foothills both to the north and to the south of the Alps. In Italy, severe hailstorms occur on all regions according to records of the National Climatic Data Center [30], but most frequently over the northern parts [31,32]. The longest historical records of hail frequency have been published for the city of Padua in the NE, extending back to the 14th century [33]. In the Trentino region, Eccel et al. [34] detected around 30 hail days per year over 35-yr observations. The Friuli-Venezia Giulia is another hail-affected region: using a hailpad network of more than 300 stations, Giaiotti et al. [35] obtained 55 hail days per year. The higher frequencies were found near Udine in the NW, with around 2.0 hail days per year, compared to values around 1.0 elsewhere (1992–2009 [29]). For a network consisting of 370 stations in Emilia-Romagna, Nanni [36] reports a mean number of 26 hail days per year (1983–1998), with a mean frequency among the observing sites of 0.7.

In Central-Eastern Europe (S3.3) with the countries Poland, Czech Republic, Slovakia, and Hungary, the frequency of severe hailstorms (SH) is most elevated in the South, where convective activity is often forced by local conditions in particularly due to the orographic influence [32].

The climate of Southern Europe (S1.2, S2.2), including Italy and the Iberian Peninsula, is dominated by the high insolation and proximity to the Mediterranean, where warm and moist air masses are advected from S to W directions. These environmental conditions are primarily responsible for the formation of severe hailstorms during late summer and autumn. As demonstrated by our analysis, the months of Sep, Oct, and Nov show the maximum occurrence of large hail (H) and super hail (SH) in the sectors covering these areas. The eastern Mediterranean (S1.2) is well known for its high exposure to hail, mainly due to the geographical situation [37–39] as investigated in the Spanish areas of Pyrenees, along the east coast and the Ebro Valley.

The environmental conditions of the central basin (S2.2) where we found the maxima of H and SH for Sep-Nov are governed by the cold fluxes from the North with moisture

gradients due to the high sea surface temperature of the Mediterranean. As shown in [30] where hail reports, claims and reanalysis are used to analyze the period 1971–2009, Southern Italy exhibits a very high frequency of hail events.

In southeastern Europe (S3.2), the general climate is influenced by the high complexity of the local conditions. The presence of the Mediterranean and the Black Sea, the coexistence of several islands, gulfs and large mountain chains (the Balkans and Anatolia), induce prominent high local variability. Our analysis shows the sensitivity of this sector both to H and SH identifying the seasonal justification of the phenomena. Although it is demonstrated [40,41] that in continental Greece hail events are not so common, maxima in hail frequency are found over the western islands of the Ionian Sea and the Eastern Aegean. High values of severe hail events were found on the Turkish Mediterranean coasts [42].

Finally, it has to be mentioned that the south-central Mediterranean, including Tunisia and Libya (S2.1), shows the same seasonality of adjacent sectors as to the maximum number of hail events. Although that region has the highest storm rates in late spring/summer, many authors ([43,44]) mention hail as a common phenomenon.

## 6. Conclusions

In this section, we discuss the possible expectation for future hailstorm frequency in the Mediterranean in the context of climate change. Although the argumentation is far from being conclusive, it represents a first step towards new studies accounting for the changes of atmospheric variables inducing changes in hail-bearing storms.

The analysis described in Section 4.2.1 demonstrates the increasing trend of H and SH events in the entire Mediterranean basin. In particular, curves in Figure 7a–c display a positive annual trend (significance > 90%) for H events following a quasi-regular growth from a minimum value of 100 H events per year (H/yr) to values higher than 300 H/yr calculated for the end of time series. As for SH, due to the intrinsic rarity of these events, the analysis suffered for the low amount of data. Thus, only time series 3 (Figure 9c) exhibits a significant positive trend (significance > 94%) with an increasing number of events from 25 SH/yr to 70 SH/yr at the end of the considered time period.

Such a steep increase of hail events during the last two decades motivated the exploration of the atmospheric variables generally considered as precursors of deep convection that can produce hailstones. Although the relation between hailstorms and the selected atmospheric variables is not direct and far from being well understood, we applied the Mann–Kendall test for evaluating the distribution of four key variables for the convective activity. Using the ERA5 reanalysis for the period 1959–2021, the annual trends of the Convective Available Potential Energy (CAPE), the Zero Degree Level (ZDEGL), the temperature at 850 hPa (T850) and the Sea Surface Temperature (SST) have been calculated for the entire Mediterranean Sea. In Figure 15, the temporal variability and trend of the selected atmospheric variables are shown. The high correspondence between our findings and graphs in Figure 15 justifies the following considerations.

The CAPE variable indicates the available energy for the development of convection. Thus, increasing CAPE values are associated with enhanced atmospheric instability and in turn with the tendency of the environment to form hail-bearing convective systems. The diagram of CAPE distribution shows almost doubling values from 1959 to 2021.

The ZDEGL height impacts on the melting of ice hydrometeors. As shown in Figure 15, during the last 62 years, the height of the melting layer increased by about 400 m. The main consequence of the increment of the ZDEGL height is an additional melting of small hail (<2 cm) before reaching the ground with a possible enhancement of liquid precipitation [45]. However, for large hail (2÷10 cm in this context) and super hail (>10 cm), an increase of kinetic energy due to the longest path of hail fall might be conceivable.

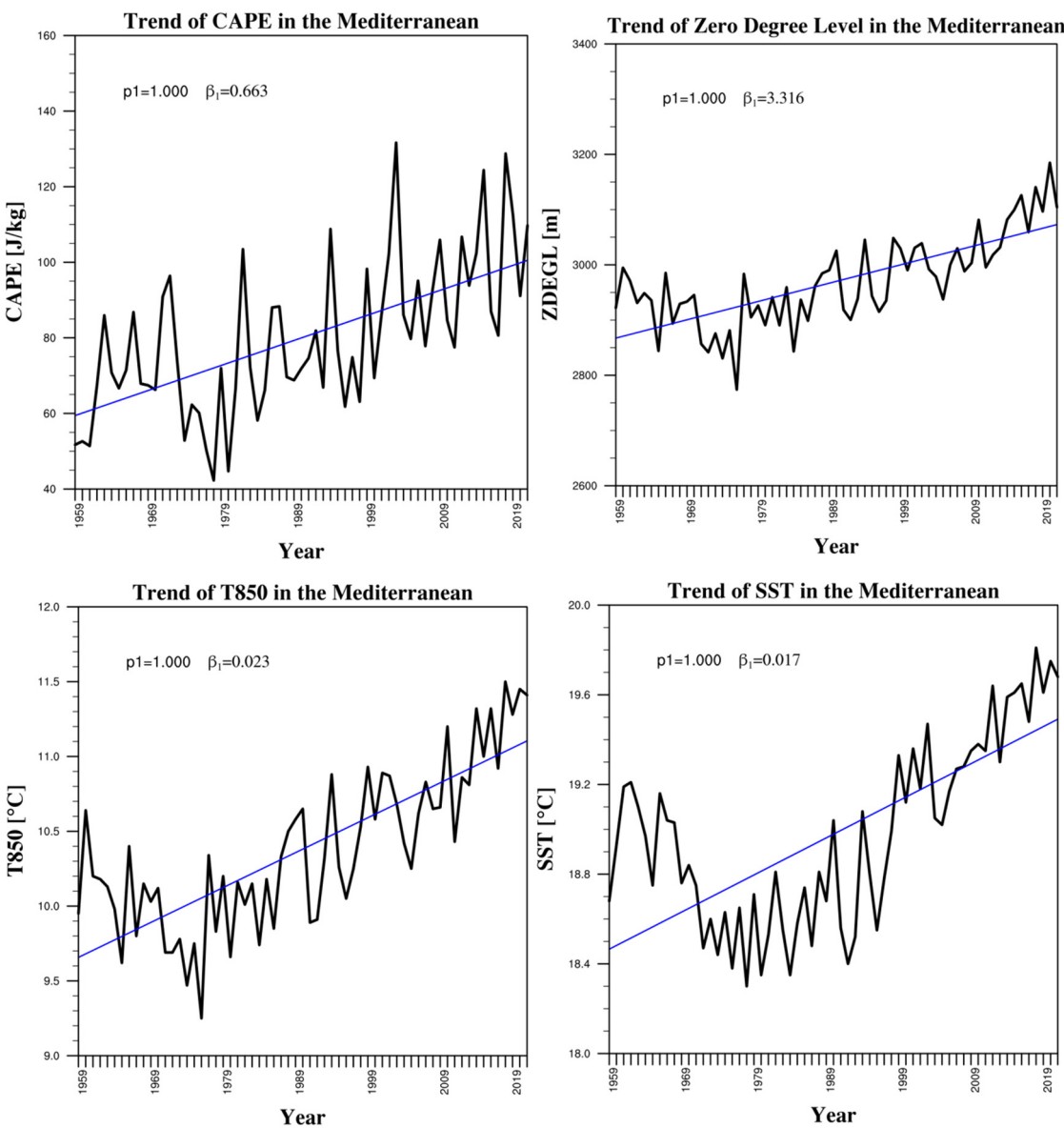

**Figure 15.** Annual trends for the Convective Available Potential Energy (CAPE), the Zero Degree Level (ZDEGL), the temperature at 850 hPa (T850) and the Sea Surface Temperature (SST) calculated for the entire Mediterranean Sea. Blue line displays the tendency of series.

The steep increase of the T850 and SST can be significant to regulate the mechanism of the triggering of vigorous convection. Variations of a few degrees of the T850/SST can alter the dynamics of vertical fluxes which sustain and reinforce deep convection. As demonstrated in [46], the variations of SST only of ±1 K drastically change the vertical velocity of the updraft by suppressing the formation of the tornadic supercell and hail-inducing deep convection.

Based on the above considerations, a series of new studies is planned to seek a relationship between hailstorm occurrence and most relevant atmospheric variables, chiefly the synoptic conditions mostly favorable to the formation and intensification of hailstorms in the Mediterranean basin. This would offer a robust tool for forecasters to quickly identify the synoptic configurations leading to the formation of hailstorms, in particular those characterized by very high severity.

**Author Contributions:** Conceptualization, S.L.; methodology, S.L. and G.M.; software, S.L. and G.M.; validation, E.C. and G.M.; formal analysis, G.M., S.L. and E.C.; data curation, G.M.; writing—original draft preparation, E.C., G.M. and S.L.; writing—review and editing, E.C., S.L., G.M. and V.L. All authors have read and agreed to the published version of the manuscript.

**Funding:** This research received no external funding.

**Data Availability Statement:** Data supporting the results of this work were freely downloaded from the NOAA Comprehensive Large Array-data Stewardship System (CLASS) archive for satellite radiometers AMSU-B/MHS. Satellite data of sensors GMI, SSMIS and ATMS were freely downloaded from the NASA Precipitation Processing System (PPS) Science Team On-Line Request Module (STORM) data archive. The climate variables displayed in Figure 15 are freely available on the Copernicus Climate Data Store (CDS).

**Conflicts of Interest:** The authors declare no conflict of interest.

## Abbreviation

| | |
|---|---|
| AMSU-B | Advanced Microwave Sounding Unit-B |
| ATMS | Advanced Technology Microwave Sounder |
| CAPE | Convective Available Potential Energy |
| ESWD | European Severe Weather Database |
| HI | Hail Initiation |
| H | Large Hail |
| HP | Hail Potential |
| IFOV | Instantaneous FOV |
| GMI | GPM Microwave Imager |
| GPM | Global Precipitation Measurement mission |
| GPM-C | Global Precipitation Measurement Constellation |
| GMP-CO | GPM Core Observatory |
| MHS | Microwave Humidity Sounder |
| MWCC | MicroWave Cloud Classification method |
| MWCC-H | MWCC-Hail method |
| NCEP | National Center for Environmental Prediction |
| SH | Super Hail |
| SSMIS | Special Sensor Microwave Imager/Sounder |
| SST | Sea Surface Temperature |
| TB | Brightness Temperature |
| TRMM | Tropical Rainfall Measurement Mission |
| ZDEGL | Zero DEGree Level |

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
