# Peer review of "Hail Climatology in the Mediterranean Basin Using the GPM Constellation (1999–2021)"

_remotesensing, doi:10.3390/rs14174320_

Round 1

Reviewer 1 Report

See attached PDF

Author Response

Remote Sensing 1863965

Hail climatology in the Mediterranean basin using the GPM constellation (1999-2021)

by S. Laviola and coauthors

Answers to Reviewer 1

We heartily thank Reviewer 1 for her/his thoughtful reading of the manuscript and for the encouraging words on the work being presented. The criticism is very useful and has allowed us to substantially improve the quality of the paper.

The following list of answers details the changes made to the manuscript and the actions taken to meet the Reviewer’s recommendations.

Major comments

- The overall writing quality of this manuscript is not quite publication worthy. There are numerous incomplete sentences due to missing or incorrect word usage, which makes it difficult for the reader to fully understand what the authors are trying to convey. I will only provide some examples in the next section. In general, I ask that the editors work with the authors to comb through the entire manuscript and make sure all sentences are structurally sound prior to publication.

Thank you for the general comment. We apologize for the imperfections characterizing the first submission. They were generated by the need for quickly submitting within a final deadline. The manuscript was completely revised by fine combing the English language and trying to modify those sentences that were not clear enough or even obscure.

Thank you indeed for pointing out flaws in various sections. Here is a list of answers.

- The authors provide two sentences on page 2 (Introduction) to defend the uniqueness of their study. They say the current study is based on two prior papers published by the first author with no additional information or context provided. The authors then suggest that they are the first to exploit satellite measurements for long term spatial variability of hail events. Hopefully this is an easy correction, but the authors must expand on this paragraph. First, the sentence “This work has its premises in the results presented in Laviola et al. [1,2]” does not provide meaningful information without additional summary. The reader should not be required to read two additional papers to determine the purpose of the current study. Please summarize the pertinent methods and/or results from Laviola [1,2] and how this current study expands on them to produce a new and publication worthy analysis.

The Reviewer is right. We added a couple of sentences highlighting the findings of the two previous papers in that enabling the reader to place the current work in the wake of these essential publications.

- I also noticed that the citations are labeled [1,2], even though this is the first sentence making mention of the Laviola papers and citations [3 – 7] were already discussed in previous paragraphs. Perhaps the authors originally included such a summary at the beginning of the paper and ultimately removed it prior to submission?

The Reviewer is right, and the numbering of the referenced papers now follows the order of first citation in the text.

- Second (and perhaps this will be clarified by adding a proper summary of the Laviola [1,2] papers), the authors state their study is unique because it is the first to exploit long term satellite retrievals, but earlier it is stated that Punge and Kunz [3] performed an exhaustive review and described the spatial variability of hail frequency over Europe with records long enough to talk about potential climate change impacts. What is different between the current study and the results provided by Punge and Kunz [3]? It was a review paper, so I assume a variety of datasets, methods, and analysis were used and summarized together. Was the goal of this study to create a European/Mediterranean hail climatology database using a single method that hadn’t yet been performed on such a long dataset before – thus providing a longer and more consistent database than had previously existed? If so, this should be mentioned and made clear because it is crucial information in the defending of the uniqueness of this study.

Yes, the Reviewer is right, and an explanation was clearly needed. Now the findings of the two papers by Laviola et al [15,16] are briefly summarized and this already helps in this direction. However, the difference between the present study and a previous relevant one cited in the Introduction is in the use of the constellation of satellite microwave radiometer frequencies. This allows to exploit cloud structure as observed by cloud-penetrating microwave frequencies with an adequate time repetition. This is now expressed in the text.

- The second sentence of the Discussions on page 15 states “The main reason that justifies our study is producing a new climatology of hail episodes affecting a very complex domain which is well-known to be a planetary climate hotspot, namely a planet region particularly vulnerable to the impact of climate change.” But the reader should not be required to reach section 5 before finally understanding the primary goal of the study (which is what happened to me as I performed this review). Please better define the purpose and defend the uniqueness/relevance of this study in the Introduction, taking these comments into account.

Yes, the Reviewer has a rather good point. The main purpose was described only in the Discussion section, i.e., at the end of the paper. Now the purpose of the study is detailed in the Introduction on page 2 where the previous studies are described (see above) and the uniqueness of the paper defended.

In summary, the paragraph defending the need for this paper now reads:

“This work has its premises in the results presented in Laviola et al. [15,16]. Laviola et al. [15] devised a new logistic model for the relation between the scattering signal in the microwaves and the hail diameter by varying the hail cross-section; the model quantifies the extinction of radiation due to hailstones and ice aggregates through the application of a probabilistic growth model. The model was applied to the satellite platforms in orbit and validated against 12 years of surface observations. The method was then extended to the Global Precipitation Measurement Constellation (GPM-C) [16] to account for instrumental differences among the various microwave radiometers and attain a global observation perspective at high temporal rate; performances were demonstrated very high in the detection of different hail-bearing storms and seasonality of the events. The present study is the first that fully exploits the potential of cloud-penetrating satellite microwave radiometer frequencies to derive the occurrences and the long-term (1999-2021) spatiotemporal variability of hail events on the Mediterranean basin based on the potential of the probabilistic method using the GPM-C. The study provides a long and consistent dataset based on physical cloud observations that was not done previously. In fact, the main reason that justifies the study is producing a new climatology of hail episodes affecting the very complex Mediterranean domain, which is well-known to be a vulnerable planetary climate hotspot.”

Minor Comments and Clarification

Thank you very much indeed for listing imperfections and asking questions that contribute to improve the text and the overall quality of the paper.

- Abstract: Even the very first sentence of the paper is grammatically incorrect. How about something like this: “The increasing interest in the studying of hailstorms and their costly impacts have resulted in several publications.”

The sentence was indeed ambiguous and not quite grammatically correct. The new sentence is:

“The impacts of hailstorm on human beings and structures and the associated high economic costs have raised a high interest in studying storm mechanisms and climatology thus producing a substantial amount of literature in the field.”

- Line 12: Authors state assessing hail frequency during 1999-2021 using retrievals derived from the GPM-C radiometers. However, GPM began in 2014 (as stated by the authors and shown in Figure 2). Please rephrase this sentence to be accurate, as multiple satellite platforms were merged in order to assess hail prior to 2014.

Yes, the GPM Core Observatory was launched in February 2014 and our analysis includes the 1999-2014 period. The sentence now reads as follows:

“To contribute to this field, we have explored the hail frequency in the Mediterranean basin during the last two decades (1999-2021) on the basis of hail occurrences derived from the observations of the GPM-C microwave radiometers from 2014 onwards and merging multiple other satellite platforms prior to the GPM launch.”

- Line 19: Abstract states a difference in trend between the time frames 2010-2021 and 1999-2010, with the latter period being “more steep”. However, in the text (section 4.2.1, for example) only mean values of hail occurrence are calculated between the two time periods, not trend lines. So what is meant by “more steep” here? Please clarify.

Trends are presented (e.g., in Fig. 8 and 9) and show that there is a noticeable increase of hail events in all categories after 2010 and the difference with the previous period is evident. The sentence has been rephrased as follows:

“The trend analysis shows that the mean values of the hail events over the entire domain tend to substantially increase, showing a higher increase during 2010-2021 than during 1999-2010.”

- Lines 21 – 22: “that possible be” ? Perhaps rephrase to “that might be”?

Changed to “that might be”.

- Introduction, paragraph 2, lines 32 – 39: what timeframe is covered by the Punge and Kunz [3] review paper as compared to the 1999-2021 period of the current study?

The review by Punge and Kunz describes various studies spanning from “recent decades” up to “more than a 100 years”. However, the studies are mostly based on ground reports and not on satellite detection of hailstorms, let alone by microwave radiometers.

- Line 44: “growing” should be “growth”. This is done multiple times throughout the paper.

Thank you for spotting this flaw. It was corrected throughout the manuscript.

- Line 44: perhaps “drawing” should be rephrased to “inferring?

Changed to “inferring”.

- Lines 49, 52, etc: “convections” should be “convection”. This is an issue throughout the paper.

Yes, correct. It has been modified throughout the whole manuscript.

- Lines 53 – 54: quantify this overestimation. By what percentage does this method overestimate hail frequency? Was this overestimate corrected for in the current study?

The presence of an overshooting top is normally a synonym of a deep severe storm, which can be associated with the presence of hailfall. However, hail production is linked to the actual physics of the storm and not all large storms produce hail during their development. The percentage of overestimation in methods that rely upon the detection of overshooting tops as a way of detecting hail on the ground is not specified in the literature. The method used in the paper does not use the detection of overshooting tops and uses a statistical-physical approach. Thus, the overshooting top presence is not used at all.

- Line 65: remove “have”

Done.

- Lines 68 – 71: See Major Comments #2

See answer to Comment #2 above.

- Lines 74 – 75: Again, this paragraph makes mention of using a hail retrieval method as applied to the GPM-C constellation. But the technique is ultimately used on other satellite datasets in this study. Please clarify. Perhaps state the retrieval method was “originally” applied to the GPM-C datasets?

Yes, a clarification is now provided and the sentence now reads:

“The hail dataset exploited in the present analysis is derived from the probability-based algorithm Micro-Wave Cloud Classification-Hail (MWCC-H) applied to frequency channels in the range 150–170 GHz of the GPM-C and modified for applications to the sensors available in the pre-GPM era.”

- Line 75: graupels should be “graupel”. This is an issue throughout the paper.

Yes, correct. It has been modified throughout the whole manuscript.

- Lines 72 – 81: the hail retrieval technique is mentioned but not described in any detail. At the same time, no citations are provided in this paragraph. How are brightness temperatures linked to hail probability (?), and please provide the proper reference(s) from where this technique originates.

The references to the paper describing the method are now included (Laviola et al. [15,16]). Describing the method in detail is not the purpose of the current paper as it has already been described in the referenced papers However, a brief sentence has been added. The whole sentence now reads:

“The hail dataset exploited in the present analysis is derived from the probability-based algorithm Micro-Wave Cloud Classification-Hail (MWCC-H) [15,16] applied to frequency channels in the range 150–170 GHz of the GPM-C and modified for applications to the sensors available in the pre-GPM era. The MWCC-H method associates probability values to microwave signals from very small ice particles (low probability) while very large hailstones (d > 10 cm) are typically marked by likelihoods close to 1.”

- Line 82: “allowed to evaluate”… please rephrase/clarify… maybe “allowed for the evaluation of”

Modified as suggested.

- Line 86: “of study area” should be “of the study area”

Done.

- Section 2, Line 109: rephrase to “identify the areas most affected by hail” or something similar

Done.

- Section 3, Line 122: “suffer for” should be “suffer from” ?

Done.

- Line 123: should just be “from 2001”

Done.

- Line 132: Table 1 doesn’t actually describe the damage caused by hail – only a general intensity range. I recommend changing “potential damages” to “potential severity”

Suggestion well taken. “Damages” was substituted with “severity” in the text and also in the Table and its caption.

- Line 133: I think “grows” should be “growth” here. Also, please clarify what is meant by “kinetic energy”. Is this kinetic energy of the storm updraft? Derived from CAPE or measured directly? Something else?

The sentence has been rephrased as “Note that the increase of the hail probability as a function of hail diameter is associated with an increase of the kinetic energy and in turn of the severity of the event.”

The kinetic energy is that of the individual stone and is computed with the formula based on the fourth power of the radius. An appropriate reference has been added:

Strong, G.S; Lozowski, E.P. An Alberta study to objectively measure hailfall intensity. Atmosphere 1977, 15, 33-53. https://doi.org/10.1080/00046973.1977.9648429

- Lines 138 – 139: “Although the importance of the four categories to fully describe the dynamics of hailstorms,” is not a complete sentence. What are the authors trying to say about the importance of the 4 categories? Please rephrase to clarify what the authors mean here.

Yes, the sentence was not clear at all. It has been rephrased as:

“The four categories are introduced to describe the characteristics of the hailstones from very small to graupel and finally large hail. These four categories are instrumental to fully describe the dynamics of hailstorms, but in this study only categories associated to the most severe events have been selected.”

- Line 142: “graupels”

Done.

- Lines 172 – 175: The method employed in this study identifies hail pixels as opposed to number of actual hailstorms (multiple pixels might correspond to the same storm, for example). Is this method consistent with other studies? Bigger but fewer hailstorms could theoretically result in the same number of pixels as smaller but more frequent hailstorms. Do the authors foresee any issues with this as related to their overall findings?

Thanks for this comment. The basic hypothesis of our analysis is the definition of “hail event”. Each “hail/superhail event” has been defined as a single pixel identified by the MWCC-H. Thus, neither the physical shape nor the geographic extension of the storms were considered, but only their magnitude according to the MWCC-H classification.

Our approach is justified on the basis of the ratio between areal extension of hailstorms and native spatial resolution of microwave sensors of the GPM-C (10-25 km). Generally, hail clouds are isolated local events rapidly evolving in space and affecting spatial domain typically within 10 km. Thus, the ground resolution of the satellite sensors used for detecting hail are 1 to 2 times wider than area usually hit by hailstorms. Therefore, due the coarse resolving power of microwave sensors with respect the spatial scale of hailstorms each pixel counted by our analysis catches all hailstorms inside the satellite pixels. Of course, the limits of our results are found in the loss of small-scale hailstorms both in terms of numbers and of intensity.

- Results, Line 212: should be “events that occurred”, or “which occurred”

Done.

- Lines 213 – 214: This sentence is confusing: “All the results discussed in this section are obtained using all satellites available during the period 1999-2021.” In section 3.2, it is stated that only one operational platform was used at a time for a given time period and that the three different combinations of time series presented in Table 2 were all analyzed to verify robustness of the results. Thus the results shown in Figure 4, for example, can’t show results from all the satellites. It should be based on only one of the time series shown in Table 2, correct? Please clarify which time series or combination of which satellites is plotted in Figures 4 – 6. How are the higher resolution satellite datasets (GMI) being included if spatial resolution consistency throughout the timeseries is required?

The three-time series reported in Table 2 were used only to verify the robustness of the statistical trends (Sect. 4.2). Figures 4 – 6 show the monthly distribution (full domain in Fig.4 and in the nine sectors in Figs. 5-6) of the hail/superhail events detected by all satellites of the GPM-C, spanning from only one sensor on the NOAA-15 satellite in 1999 (AMSU-B) to four sensors (MHS, ATMS, SSMIS, GMI) equipping different platforms in 2021.

The MWCC-H adjusted to all MHS-like radiometers of the GPM-C, namely ATMS, SSMIS, GMI (see Laviola et al., 2020b), preserves the native resolution of each sensor. Thus, the conical scanning such as GMI and SSMIS resolve IFOVs at 8 and 14 km, respectively, while cross-tracks (AMSU-B/MHS/ATMS) are definitely worse by observing ground-swaths in the range 16-48 km.

In this study, GMI and SSMIS were exploited with their native spatial resolution but for the cross-track sensors we considered only IFOVs close the nadir position by preserving the spatial resolution in the range 16-25 km (20 IFOVs at edges of scanline were rejected). This is specified in the text.

- Line 226: “that” or “which occurred”

Done.

- Line 227: “Figure” should be “Figures”

Done.

- Line 228: In what way do Figs. 5 and 6 “clearly support” the results of Fig. 4? As described below, many of the sectors show hail peaks at different times of year compared to the overall time series of Fig. 4

The general overview offered by Fig.4 shows the seasonality of H and SH events occurred over the whole Mediterranean Basin without specifications of the position (sectors) of maxima. Figs. 5-6 specify where those maxima are located. The coherence between figures is found by comparing the position of two peaks of Fig. 4 with the color bars of Figs. 5-6. The first peak of Fig. 4 (Jun-Aug) corresponds to sectors 2.3 and 3.3 of Fig. 5 and the second peak (Sep-Oct) with sectors 2.2 and 3.2 of Figs. 5-6.

The original sentence is now rephrased as “The monthly occurrence of the H and SH events in each sector is coherent with the distribution of hail calculated for the whole Mediterranean basin (Fig. 4), but provides also further insights concerning the spatial distribution of the hail seasonality.”

- Line 249: should be “protected by a subtropical…”

Done.

- Line 285: should be “in the case”.

Done.

- Line 288: The authors split the time series into subseries 1999 – 2010 and 2010 – 2021. What was the authors’ reasoning behind including the year 2010 in both series? And how do the authors balance the desire to split the series into two equal datasets (in time) even though the “split-trend” being discussed occurs closer to 2013?

The year 2010 has been included in both series to create two symmetric datasets in order to calculate a more consistent trend analysis on the basis of two homogeneous dataset in terms number of years.  

The “split-trend” observed closer to 2013 should be attributable only to accidental motivations, depending marginally on the time series considered.

- Line 298: Is it common to use the phrase “24 UTC”? Should this “18-00 UTC”? This also occurs multiple times throughout the remainder of the paper.

The reviewer is right. It is better to use “0000” instead of “2400”. The change was made throughout the text and the hours were indicated as 0000, 0600, 1200 and 1800 to be more consistent and clearer.

- Lines 352 – 353: Please rewrite this sentence – its meaning is difficult to discern. What is meant by saying the similarity among events is scarce? Are the authors attempting to say that the three time series are not similar? Or are the authors discussing the scarcity of observations (small numbers of H pixels)?

The sentence was rephrased as “However, the similarity among the three time series of the H events is very low.”

- Lines 363 – 364: Why is the Libyan sector considered questionable but not the southeastern Mediterranean? The number of H events appears to be significantly lower in Fig. 10 than Fig. 11.

Basically, the hail frequency in southeastern Mediterranean could be much lower than in the northern sectors. Nevertheless, it happens that a significant event may occur from time to time. However, “questionable” is not the proper word and was modified.

- Line 366: Authors state “Gargantuan” hail but provide no citation/reference.

Yes, “Gargantuan” was modified in “very large”.

- Line 370: should this sentence read “significant positive trends” ?

Yes, modified as suggested.

- Section 4.3: Why were the years 2013 – 2018 chosen? Did this produce a similar number of analyzed pixels in each diurnal time period?

Yes, this is the main reason for choosing the years 2013-2018.

- Line 376: “explore which is the portion”… please rewrite

Modified as “”to identify the time of the day more affected by…”

- Lines 381 – 384: Are these platform combinations still consistent in spatial resolution?

Yes.

- Line 386: should be events “that” or “which” occurred

Done.

- Line 396: Previous studies state hail events are more common in the afternoon, and the authors suggest that a similar result was found with their super hail analysis but not hail analysis. But figure 14 shows peak frequencies for H and SH during the 18 – 24 UTC period. This is evening in both cases (not afternoon), correct? Am I missing something here?

Sorry, problem with nomenclature. Afternoon is actually “late afternoon”. However, “evening” is much more correct.

- Line 411: what is a “planet region”?

Right, “planet” is wrong and was removed leaving just “region”.

- Lines 432 – 433: please reference some of these “several” studies

It is sufficient to reference Punge et al. [1]. The sentence is now rephrased: “Central Europe (S1.3, S2.3) is highly exposed to hail hazard, namely Switzerland, Northern Italy, Austria, and Germany, where several comprehensive studies on hail frequency are available (see [1] for a comprehensive review).”

- Lines 446 – 447: “were found” is typed twice

Corrected.

- Line 450: What is meant by a mean point frequency of 0.7 and what is its physical significance?

“Point” would eventually mean “local” in terms of a very small spot, as opposed to “area”. It is better to omit “point”. The sentence is rephrased to:

“For a network consisting of 370 stations in Emilia-Romagna, Nanni [36] reports a mean number of 26 hail days per year (1983-1998), with a mean frequency among the observing sites of 0.7.”

- Line 458: “the main” should be replaced by something like “primarily”, and “responsible of” should be “responsible for”

Done.

- Line 474: what is meant by “seasonal justification”? Should this be “variability”?

Done.

- Lines 480 - 481: These sentences do not make sense. Should “showing” be “show” or “exhibit”? What is meant by “Maximum values of hail events.” ?

The sentence is now rephrased as “Finally, it has to be mentioned that the South-Central Mediterranean including Tunisia and Libya (S2.1) shows the same seasonality of adjacent sectors as to the maximum number of hail events.”

- Lines 483 – 522: First, why is this section and Figure 15 presented in the Discussion? Shouldn’t the analysis be presented in Section 4 and simply summarized in section 5?

We thought of presenting separately the graphs in the “Discussions” section in order to catch the attention of the reader on the climatic context in which our results are framed. The graphs are instrumental to show the general variations of the Mediterranean environment during the last decades. Thus, if on one hand the trend of Fig. 15 clarifies the behavior of the main climatic variables linked to the intensity of convection, on the other they reinforce our results presented for the Sect. 4.2.

- Line 483: “on” should be “the”

Done.

- Line 489: “grow” should be “growth”

Done.

- Line 494: should be “at the end”

Done.

- ERA 5 analysis, starting at line 500: Did the authors consider or account for any inconsistency between the ERA 5 and satellite derived hail datasets? Specifically, the hail datasets were only derived using the months April – November. Yet Fig. 15 shows trends in “annual” mean variables. Do the authors expect the same trends if only mean April – November mean trends were shown?

We computed the new trends for the period April-November. No variation is detected with respect to the previous ones (increasing trends): the confidence level is identical.

- Line 500: Perhaps clarify why the start year of 1959 was chosen even though the associated satellite datasets began at 1999.

The start year of 1959 was chosen to stress the significant increasing trend over a more than 60-yr time series.

- Line 503: Clarification - what is meant by the “entire Mediterranean Sea”? Is this truly an analysis over only the sea for every variable displayed in Fig. 15? Or was CAPE, for example, calculated over the entirety of the domain described in Fig. 1? Shouldn’t these variables (including surface temperature, not just SST) be calculated over the land masses where the hailstorms occur and pose threats to populations at the surface?

All ERA5 variables were calculated and averaged on the entire domain (land/sea) as displayed in Fig. 1. Only SST as obviously averaged over the water domain.

- Line 514: division sign should be “-“ ?

The sign is “÷”, meaning from 2 to 20 cm.

- Lines 515 – 516: This seems like a lot of speculation and is not clear: “an increasing of kinetic energy due to the longest pathway of hailfall might to be conceivable.” Please clarify what is meant by this sentence and rephrase. How is kinetic energy increased with a higher freezing level? Are the authors speculating that for large hail, greater melting and evaporation during fallout leads to greater cooling and negative buoyancy imparted on the falling hail? What does this have to do with hail frequency as observed by satellites, which was the overall focus of the study?

No, the sentence is perhaps not clear, but the increase of kinetic energy refers to the longest path covered by the hailstones while falling. The stones increase their size and thus the kinetic energy increases. The sentence was rephrased as “However, for large hail (2÷10 cm in this context) and super hail (> 10 cm in the context of the work) an increase of kinetic energy due to the longest path of hail fall might be conceivable.”

- Lines 517 – 522: I understand that the authors reference a paper [47] that discusses SST variability on potential updraft strength. However, the combination of 850-mb temperature and SST analysis appear to more closely relate to the low-level lapse rate impact on convective updraft intensity. Why not simply plot and present the lapse rate change over the last 60 years? I assume that reference [47] was changing SST based on a constant T850 above and ultimately discussing supercell and hailstorm suppression based on the resulting lapse rate impacts on convective strength?

Yes, the Reviewer is right. However, diagrams in Fig. 15 just give an overview on the progression of the changes over the Mediterranean during the last 60 years. That’s why a non-exhaustive formulation of the “Discussions” section was implemented. 

The reference [47] was included not for discussing the suppression of the tornadic supercells but just for showing the potential of SST in the intensification of convection usually associated with the formation of hailstones.

- Line 523: I assume “latter” should be replaced by “prior” or something similar?

Done.

- Lines 523 – 533: This section is not necessary. If the authors wish to keep it, it should at least be written more concisely for clarity and improved sentence structure.

The paragraph has been rephrased as “Taking advantage from prior considerations, a series of new studies has been planned to seek a relationship between hailstorm occurrence and most relevant atmospheric variables, chiefly the synoptic conditions mostly favorable to the formation and intensification of hailstorms in the Mediterranean basin. This would offer a robust tool for forecasters to quickly identify the synoptic configurations leading to the formation of hailstorms, in particular those characterized by very high severity.”

Reviewer 2 Report

I just deplore that you do not present any correlation between satellite and ground based measurements... That could be presented in one figure

Author Response

Remote Sensing 1863965

Hail climatology in the Mediterranean basin using the GPM constellation (1999-2021)

by S. Laviola and coauthors

Answers to Reviewer 2

Comments and Suggestions for Authors

I just deplore that you do not present any correlation between satellite and ground based measurements... That could be presented in one figure

We thank the Reviewer for the comment on this important aspect. In general, ground-based hail detectors (i.e. hailpads) are often absent or unevenly distributed. Therefore, hail datasets frequently suffer for the limited spatial distribution and/or irregular acquisition time. The method used in this study for producing the hail climatology has been validated using 12 years of data (March through September) from surface hail observations over the US (NOAA-SPC storm reports) collocated with AMSU-B/MHS overpasses. Each AMSU-B/MHS overpass was combined with hail occurrences at the ground within a 1° grid box lat/lon and represents the total number of hailstorms over the entire spatial domain. Further details can be found in Laviola et al. [15]. That analysis has demonstrated the high performance of the hail detection method (MWCC-H) in distinguishing between moderate and severe hailstorms, fitting the seasonality of hail patterns. The flexibility of the method allowed for migrating its performances to other microwave radiometers of the GPM constellation (Laviola et al. [16]). Thus, the statistics presented in this work was calculated based on this fundamental heritage.

Reviewer 3 Report

The authors of the paper exploited multiple satellite platforms and a probability-based algorithm to extract hailstorm probability from passive microwave data.

The analysis was carried along a two-decade time series, and for the purpose of demonstrating a monotonic trend in the hailstorm probabilities, three separate time series were assembled using data from different satellite platforms. Since the increase of data availability would naturally result in an increase in the number of detected hail events, the authors built the time series by assembling sensors with the same spatial resolution while still covering the entire period of the analysis. By keeping only one platform operational at a time it was possible to maintain evenly spaced samples all throughout the analyzed time period, thus making identified trends significant.

The chosen study area incorporates almost the entire Mediterranean basin, and was divided into nine sections. The choice is extensively justified in chapter 2, and the division in nine sections was not only utilitarian, but also allowed the authors to identify trends and discuss them in relation to the local characteristics.

In general, the approach used by the authors is sound and clearly explained all throughout the paper. The discussion of derived the data is always followed by a phenomenological explanation and often also by a comparison with the existing literature, even when the latter does not fully align with the authors' findings.

In the reviewer's opinion, the paper can be accepted for publication, provided the inclusion of a small number of revisions and handful of suggestions and corrections relating to the usage of the English language.

Line 9: "resulted to" should be corrected to resulted "in"
Lines 21-22: the final phrase makes little sense in English: "namely Mediterraneans regions most susceptible to the hail events that possible be more vulnerable to the future changes of climate" should probably be corrected to "namely Mediterranean regions most susceptible to the hail events that could possibly be more vulnerable to the effects of climate change" or something similar.
Line 123: "from the 2001" should be "from 2001"
Line 133: the usage of the construct "the more X increases, the more Y increases" is weird. The phrase could be rewritten as: "the higher the hail probability as a function of the hail diameter, the higher is the associated kinetic energy and the severity of the event"
Line 138: "although" does not seem the correct adverb to start this phrase. If the authors want to remark that despite excluding the two bottom categories of hail, the possibility to describe them all is still very important, they could maybe start with "Without diminishing the importance of..."
Lines 196-197: since I did not fully understand what the authors meant by "without missing values", I checked the trend_manken function documentation and found out that it requires "evenly spaced values". I suggest the authors to conform to this wording in order to avoid confusion.
Line 412: "susceptive to" should be "susceptible to"
Line 425: "the climate change" should be just "climate change"
Line 446: remove the repetition of "were found"
Line 458: "the main responsible of" should be "the main responsible for"
Line 468: "from the Southern Italy" should be "from Southern Italy"
Line 471: "the presence of Mediterranean" should be "the presence of the Mediterranean"
Line 475: "is not so common" should be "are not so common", since it is referred to hail events
Lines 479-482: the entire period is unclear, and probably underwent a couple of rewrites that resulted even in a small phrase without a verb "Maximum values of hail events". The authors should take a closer look to this part and rewrite it more clearly.
Line 489: "a quasi-regular grow" should be "a quasi-regular growth"
Line 523: "taking advantage from" should be "taking advantage of"
Line 528: since the authors are enumerating future lines of research, they should switch to the future tense: "we will reconstruct"
Lines 530-531: "from one hand/from the other" should be "on one hand/on the other"

Author Response

Remote Sensing 1863965

Hail climatology in the Mediterranean basin using the GPM constellation (1999-2021)

by S. Laviola and coauthors

Answers to Reviewer 3

Comments and Suggestions for Authors

The authors of the paper exploited multiple satellite platforms and a probability-based algorithm to extract hailstorm probability from passive microwave data.

The analysis was carried along a two-decade time series, and for the purpose of demonstrating a monotonic trend in the hailstorm probabilities, three separate time series were assembled using data from different satellite platforms. Since the increase of data availability would naturally result in an increase in the number of detected hail events, the authors built the time series by assembling sensors with the same spatial resolution while still covering the entire period of the analysis. By keeping only one platform operational at a time it was possible to maintain evenly spaced samples all throughout the analyzed time period, thus making identified trends significant.

The chosen study area incorporates almost the entire Mediterranean basin, and was divided into nine sections. The choice is extensively justified in chapter 2, and the division in nine sections was not only utilitarian, but also allowed the authors to identify trends and discuss them in relation to the local characteristics.

In general, the approach used by the authors is sound and clearly explained all throughout the paper. The discussion of derived the data is always followed by a phenomenological explanation and often also by a comparison with the existing literature, even when the latter does not fully align with the authors' findings.

In the reviewer's opinion, the paper can be accepted for publication, provided the inclusion of a small number of revisions and handful of suggestions and corrections relating to the usage of the English language.

The authors are grateful for the thorough analysis of the paper and its scientific/technical content. Gratitude is also expressed for the very positive comments on the work.

Answers to specific comments

The author wish to thank the reviewer for his/her suggestions that considerably improved the text quality.

Line 9: "resulted to" should be corrected to resulted "in"

The sentence was modified and the problem is now removed.

Lines 21-22: the final phrase makes little sense in English: "namely Mediterraneans regions most susceptible to the hail events that possible be more vulnerable to the future changes of climate" should probably be corrected to "namely Mediterranean regions most susceptible to the hail events that could possibly be more vulnerable to the effects of climate change" or something similar.

Done. The sentence was modified accordingly.

Line 123: "from the 2001" should be "from 2001"

Done.

Line 133: the usage of the construct "the more X increases, the more Y increases" is weird. The phrase could be rewritten as: "the higher the hail probability as a function of the hail diameter, the higher is the associated kinetic energy and the severity of the event"

The sentence was deeply modified also in response to another Reviewer’s comment.

Line 138: "although" does not seem the correct adverb to start this phrase. If the authors want to remark that despite excluding the two bottom categories of hail, the possibility to describe them all is still very important, they could maybe start with "Without diminishing the importance of..."

The sentence was deeply modified also in response to another Reviewer’s comment.

Lines 196-197: since I did not fully understand what the authors meant by "without missing values", I checked the trend_manken function documentation and found out that it requires "evenly spaced values". I suggest the authors to conform to this wording in order to avoid confusion.

The sentence was modified accordingly.

Line 412: "susceptive to" should be "susceptible to"

Done.

Line 425: "the climate change" should be just "climate change"

Done.

Line 446: remove the repetition of "were found"

Done.

Line 458: "the main responsible of" should be "the main responsible for"

Done.

Line 468: "from the Southern Italy" should be "from Southern Italy"

Done.

Line 471: "the presence of Mediterranean" should be "the presence of the Mediterranean"

Done.

Line 475: "is not so common" should be "are not so common", since it is referred to hail events

Done.

Lines 479-482: the entire period is unclear, and probably underwent a couple of rewrites that resulted even in a small phrase without a verb "Maximum values of hail events". The authors should take a closer look to this part and rewrite it more clearly.

Yes, the Reviewer is right. The sentence was modified and reads as:

“Finally, it has to be mentioned that the South-Central Mediterranean including Tunisia and Libya (S2.1) shows the same seasonality of adjacent sectors as to the maximum number of hail events. Although that region has the highest storm rates in late spring/summer, many authors ([43,44]) mention hail as a common phenomenon.”

Line 489: "a quasi-regular grow" should be "a quasi-regular growth"

Done.

Line 523: "taking advantage from" should be "taking advantage of"

The sentence was modified as follows: “Based on the above considerations, a series of new studies is planned to seek a relationship between hailstorm occurrence and most relevant atmospheric variables, chiefly the synoptic conditions mostly favorable to the formation and intensification of hailstorms in the Mediterranean basin.”

Line 528: since the authors are enumerating future lines of research, they should switch to the future tense: "we will reconstruct"

The entire paragraph was shortened and rewritten.

Lines 530-531: "from one hand/from the other" should be "on one hand/on the other"

The sentence was modified. The entire paragraph now reads as “Based on the above considerations, a series of new studies is planned to seek a relationship between hailstorm occurrence and most relevant atmospheric variables, chiefly the synoptic conditions mostly favorable to the formation and intensification of hailstorms in the Mediterranean basin. This would offer a robust tool for forecasters to quickly identify the synoptic configurations leading to the formation of hailstorms, in particular those characterized by very high severity.”
